# Developing and Planning a Protocol for Implementing Health Promoting Animal Assisted Interventions (AAI) in a Tertiary Health Setting

**DOI:** 10.3390/ijerph20186780

**Published:** 2023-09-18

**Authors:** M. Anne Hamilton-Bruce, Janette Young, Carmel Nottle, Susan J. Hazel, Austin G. Milton, Sonya McDowall, Ben Mani, Simon Koblar

**Affiliations:** 1Stroke Research Programme, Basil Hetzel Institute, The Queen Elizabeth Hospital, Central Adelaide Local Health Network, Woodville South 5011, Australia; anne.hamilton-bruce@sa.gov.au; 2Allied Health and Human Performance, University of South Australia, Adelaide 5000, Australia; janette.young@unisa.edu.au (J.Y.); carmel.nottle@unisa.edu.au (C.N.);; 3School of Animal and Veterinary Science, The University of Adelaide, Roseworthy Campus, Roseworthy 5371, Australia; susan.hazel@adelaide.edu.au; 4Adelaide Medical School, The University of Adelaide, Adelaide 5005, Australia; austin.milton@adelaide.edu.au (A.G.M.); simon.koblar@adelaide.edu.au (S.K.)

**Keywords:** Animal Assisted Interventions, health service organization, action research, human welfare, animal welfare, One Health/Welfare

## Abstract

The Ottawa Charter identifies that multiple levels of government, non-government, community, and other organizations should work together to facilitate health promotion, including in acute settings such as hospitals. We outline a method and protocol to achieve this, namely an Action Research (AR) framework for an Animal Assisted Intervention (AAI) in a tertiary health setting. Dogs Offering Support after Stroke (DOgSS) is an AR study at a major tertiary referral hospital. AAI has been reported to improve mood and quality of life for patients in hospitals. Our project objectives included applying for funding, developing a hospital dog visiting Action Research project, and, subsequent to ethics and governance approvals and finance, undertaking and reporting on the Action Research findings. The Action Research project aimed to investigate whether AAI (dog-visiting) makes a difference to the expressed mood of stroke patients and their informal supports (visiting carers/family/friends), and also the impact these visits have on hospital staff and volunteers, as well as the dog handler and dog involved. We provide our protocol for project management and operations, setting out how the project is conducted from conception to assess human and animal wellbeing and assist subsequent decision-making about introducing dog-visiting to the Stroke Unit. The protocol can be used or adapted by other organizations to try to avoid pitfalls and support health promotion in one of the five important action areas of the Ottawa Charter, namely that of reorienting health services.

## 1. Introduction

The Ottawa Charter encourages multiple levels of government and organizations to work together to reorient the focus of health care from predominantly illness and disease responses towards a more holistic understanding of health that promotes quality of life and wellbeing even in tertiary and acute level settings [1]. Animals have long been engaged in healthcare [2,3,4,5,6,7], requiring multidisciplinary collaboration. Structured inclusion of such services may be one way of contributing to the reorientation of healthcare systems, one of the five important action areas of the Ottawa Charter [1]. A recent concept, that of ‘One Welfare’, emphasizes connections between human wellbeing, animal welfare, and the environment [8]. Multidisciplinary professionals across human and animal fields can work together to achieve common goals, thereby improving human and animal wellbeing [9,10]. This paper outlines a research project protocol for systematic animal-inclusive AAI in a tertiary acute hospital setting.

### 1.1. Background

The use of AAI in acute settings is an increasingly accepted practice internationally [11,12]. Studies exploring the impacts of AAI have shown that the presence of an animal can have a calming effect on some people, such as lowering blood pressure and heart rates [13,14,15,16,17,18,19]. Other research has demonstrated that AAI can improve the mental and emotional health of individuals in hospital settings as well as impact on the wellbeing of surrounding companions of patients and hospital staff [9,10,20,21,22]. However, the usefulness reported varies, which needs to be considered and addressed [23,24,25], and the impact on the non-human animal needs to be considered too [4,26,27]. Biological and physiological markers and behavior in animals can be monitored for stress, although these also have their limitations. Examples of biomarkers include cortisol [28,29,30,31,32,33,34,35,36], oxytocin, and IgA [14,15,37,38,39,40]; physiological biomarkers, for example, heart rate [32,41,42] and dog behavioral monitoring [30,31,32,33,43]. Other factors, such as the handler as well as the recruitment and selection of the dog, can affect the stress responses of the dog [26]. While understandings of the experiences of animals involved in AAI are complicated by limitations such as the heterogeneity of programs, small sample sizes, and methodological limits [4,43,44], the increasing awareness (including legislative inclusion, e.g., Australia, New Zealand, Sweden, France) of animals as sentient (feeling, sensing) beings requires that we cannot continue to treat animals as simply tools to meet human wellbeing [45,46,47,48]. It is, therefore, important to monitor for signs that may indicate stress within animals participating in AAI programs in order to adjust future interventions to account for the wellbeing of the animals involved as well [4,26,36,42].

It should be noted that in the burgeoning AAI field, the terminology describing animals that work in roles that support people can be confusing for all involved—two terms can mean the same thing or the same term can be used to describe differing roles. Here, our working definitions are drawn from Defining Terms Used for Animals Working in Support Roles for People with Support Needs [45]. Our working definition, following Howell et al. [45], is that the AAI we are researching will use visitation/visiting animals—these are dogs that improve the general quality of life in various settings (e.g., hospitals), training standards are high, but as non-assistance animals they have no right under legislation to public access (i.e., having the legal right to enter public places such as banks, restaurants and national parks, usually considered off-limits to animals due to legal regulations in that jurisdiction) [45]. The animal–handler team is well trained and primarily provides the service on a non-professional/volunteer basis. The programs are not structured, and there are no specific therapeutic goals, but some participants may experience wellbeing effects. These visiting dogs may also be therapy dogs in a therapeutic setting, where, in contrast, the use of a therapy animal aims to improve specific therapy outcomes; training standards are high, and again, there is no right to public access. The animal is integrated into structured, goal-directed therapy/treatment, overseen by a relevantly trained and licensed therapy healthcare professional.

Qualitative research has highlighted that patients who experience a stroke feel major disruptions to their lives, describing a disconnect from the person they were prior to the event and concerns about their ability to perform previous roles [49]. Furthermore, patients have described that healthcare providers focus too heavily on biomedical recovery, with little recognition of the social and emotional impacts of stroke [49]. In addition to complex emotional experiences, many patients felt anger and frustration after their stroke due to loss of control [50,51]. While a diverse range of medical and allied health interventions [11,52] have been shown to benefit recovery, these may be impeded by the emotional experiences of patients. Finding a means of reducing the inherent emotional distress associated with post-stroke experiences [53,54] is crucial.

Experiencing a stroke often has a major impact on a range of aspects of a person’s functioning. The Australian Clinical Guidelines for Stroke Management offers recommendations for treatments provided to patients in the rehabilitation stage [55]. The recommendations involve aspects of physical, functional, and cognitive rehabilitation [55]. While these are important areas for healthcare professionals to attend to in the care of stroke patients, they fail to mention potential treatment for the emotional needs of the patient. Patients may experience significant emotional impacts, including those associated with long periods away from home due to hospital stays [53]. Patients have been identified as experiencing high levels of emotional distress in acute settings [53], in addition to the mood-altering impacts that strokes can engender [56]. This may result in reduced interactions with family and friends and an additional increase in emotional distress, further hindering patient recovery [11,52]. The emotional impact of stroke reaches beyond the patient themselves. Visitors of patients in acute settings have been identified as experiencing high levels of emotional distress [54]. Furthermore, due to the nature of their work, hospital staff working in stroke care are also susceptible to burnout [57]. Based on this background literature on AAI, as well as AAI in hospitals and for stroke and the impact on humans, this project aims to extend understanding of both human and animal experiences in the context of AAI.

### 1.2. Research Protocol Aims

By using Action Research at a major hospital, the Dogs Offering Support after Stroke (DOgSS) study aims to find out not only if dog visits make a difference to the expressed mood of stroke patients and their informal supports (visiting carers/family/friends), but also the impact these visits have on hospital staff and volunteers, the dog handler and dog involved, as the wellbeing of all needs to be considered to be of benefit to the community. The purpose is to provide the opportunity for research translation through embedding the incorporation of AAI as a means of health promotion in this acute care setting, assisting in the reorientation of this health service as desired in the Ottawa Charter. However, before such research can be undertaken, the project needs to be planned, a team of researcher investigators (researchers) identified, funding needs to be acquired, and once acquired, ethics, governance, and finance administration addressed. Our protocol for project management and operations sets out how the project, from conception to initial implementation of the Action Research (AR), has been undertaken and can inform and facilitate similar projects in hospital settings.

## 2. Materials and Methods

The process for ‘on the business’, namely project management to undertake AR, and ‘in the business’, describing the operational part of the research, is described. We also summarize activities undertaken prior to and surrounding the funding application in order to be prepared when a call for applications is made and appreciate some of the activities related to submitting the application (Section A.1).

Human research ethics approval was obtained. However, informed consent was not required for the work reported in this paper as it has been developed by the research team and does not include human participants.

### 2.1. Project Management

#### 2.1.1. ‘On the Business’

‘On the business’ project management addresses ethics, governance, and finance matters in Phase 1 and ongoing as relevant (during Phases 2 and 3 into project wrap-up):Ethics includes Human Research Ethics Application to the Central Local Adelaide Health Network (CALHN), which the Royal Adelaide Hospital (RAH) is a part of, and Animal Research Ethics Application to the University of Adelaide (UoA) for approval to undertake the research, as well as notification to relevant organizations of approval, and ongoing project reporting, as well as final reporting.Governance includes:
Contractual arrangements (agreements, variations for changes, intellectual property, etc.) between the administering organization and the funder, co-researcher organizations, and service provider(s). These arrangements can be for bipartite, tripartite, or more research/service/other agreements as relevant to the project. Additionally, organizations may need to have umbrella arrangements, e.g., a memorandum of understanding between the organizations, which are separate from individual research projects. Governance undertakings also include confidentiality deeds, conduct and conflict of interest declarations, national police checks, and organizational risk management.Site-specific application (with respect to where the research will take place on the ground) for approval.Sourcing, appointing, and inducting project manager(s) (PM(s)), staff, volunteers, collaborators, students (if the Chief Investigators (CIs) agree to include students), and service providers per the project proposal and relevant contractual and organizational requirements.Other requirements include setting up an advisory panel, the Project Reference/Coordination Group (PRCG), and drafting its Terms of Reference (TOR), as well as answering media requests, attending events, and addressing necessary organizational requirements.Finance includes a cost-center application for approval for funding to be received and administered, with subsequent cost-center maintenance until the project is completed and the funds used or returned to the funder, as well as managing and monitoring the funds—funds transfers, invoicing, approvals, etc., in the interim.

The PRCG TOR includes the context, purpose, membership, and process, i.e.,:Context—the DOgSS project uses an Action Research framework to facilitate the core aims of the project to identify (1) if AAI (dog visits) make a difference to the expressed mood and self-reported wellbeing of self-selecting patients engaging with the dog, (2) the impact of (dog visits) on hospital Stroke Unit staff and volunteers, and informal patient supports; and (3) the impact of visits on the visiting dog.Purpose—of the PRCG is to oversee the core process of the DOgSS project, enabling all the key organizational and unit partners to connect with each other across the life of the project to ensure the plan of each cycle is organized and implemented as planned, troubleshoot, reflect and amend the operations of the project as per an Action Research approach, i.e., after each cycle.Membership—all Chief Investigators (CIs), Associate Investigators (AIs), Stroke Ward representatives, and Volunteer Unit representatives.Process—Meetings will occur at five key points: prior to the commencement of the project and after each cycle (3 cycles) with a final wrap-up meeting as indicated 2/3 months after the conclusion of the final cycle and its meeting.

#### 2.1.2. ‘In the Business’: Research Project Operations

This includes Action Research-related meetings, arrangements, and communications, developing worksheets for Phase 2 workflow facilitation, e.g., a workflow diagram (Section A.2) and a running sheet (Section A.3) covering the practicalities of the Phase 2 research action for each dog visit in the hospital. Research conformity with hospital nursing ward requirements and hospital animal procedures need to be checked and monitored, and hospital security must be alerted to the Action Research, its timing, activities, people involved, and permission obtained for their agreement as well.

## 3. Action Research

### 3.1. Methodological Approach and Rationale for Choice of Method

This research project is undertaken using Action Research methodology. Action Research is research that combines knowledge collection with knowledge implementation through reflective cycles, allowing researchers to collect and analyze data, which then informs further actions [58]. The project aims to take place across three reflective cycles, with data being obtained from each stage, then reflecting on and assessing the data, leading to the implementation of adjustments as indicated, building on previous actions to enhance understanding. Wellbeing data (Likert Scale [52] and Pick-a-Mood smiley faces [59]) will be collected from stroke patients receiving AAI (dog visiting), their informal supports (e.g., carers/family/friends) visiting at the time of the AAI, relevant Stroke Unit staff and volunteers. Within a busy ward setting with patients with potential communication challenges, the Likert Scale [52] and Pick-a-Mood tools [59] are deemed most appropriate as simple non-language tools for this patient group, some of whom may have aphasia. Saliva samples for biomarkers (cortisol and IgA) [35,36,37,38,39,40] will be collected from the dog before and after each ward visit, and an ethogram (behavior assessment tool) [60] will be completed at each patient visit to provide animal-specific wellbeing data. Participating patients and the dog will be able to act as their own comparative group through pre- and post-visit testing.

### 3.2. Inclusion and Exclusion Criteria

Inclusion criteria for patients who have had a stroke and are in the Stroke Unit on the Stroke Ward at the RAH, as considered by nursing staff, include any conscious, stable patient who has had a stroke and is safe (no open wounds, etc.) to be visited by the dog and handler on the Stroke Unit, competent to give informed consent and who expresses an interest in participating and being visited by a dog and handler. Exclusion criteria include stroke patients who do not want to be visited by the dog and handler and/or with whom investigators cannot communicate in English (no interpreter will be used in this study). Any patient who appears to be upset or has a cognitive impairment (other than mild), intellectual disability (with significant past history), or diagnosed mental illness (other than mild depression) will be excluded.

These criteria determine eligibility for participation in this study, which will be determined by the nursing staff within the Stroke Unit. This process will occur prior to contact with the visiting dog and will be repeated for each dog visit within each cycle. Consultation was undertaken previously with the staff (nursing and medical) of the Stroke Unit and hospital Volunteer Services Unit volunteers, fitting with the Action Research stage of planning implementation, including negotiating the site and permissions. Informal supports, staff, volunteers, and the dog handler will be given an open invitation at the time to provide anonymous comments regarding their experience on a form (Section A.4) to be placed (when completed) in a sealed anonymous Comments Box on the Stroke Unit, for collection at the end of each day of visiting until the end of the cycle.

### 3.3. Sample Size

Up to 80 patients in total (approximately 20 patients/cycle, although this may be more, depending on the number of patients seen per day and consideration of the welfare of the dog × 3 cycles plus the opportunity for up to a further 20 new or repeat visits to longer-term stay patients, the latter flexibility allowing accommodation of repeat requests for patients still on the ward and the opportunity for assessment with respect to such repeat visits, which could also impact on subsequent organizational decision-making. One dog will be used throughout for consistency, if possible, but with allowance for up to three different dogs, if necessary. Our Action Research uses convenience sampling of as many patients as we can consent for Cycle 1 as a pilot for power calculations for Cycle 2. As the RAH receives approximately 1200 stroke patients per annum, we anticipate being able to recruit the stated number of patients. Informal support, staff, and volunteer numbers will vary due to the numbers present at the time of and involved with, the dog visit.

### 3.4. Participant Recruitment Strategies, Timeframes and Data Collection

DOgSS was planned as a 24-month Action Research project including three phases (Phase 1—project implementation, Phase 2—data collection, Phase 3—analysis and write-up) [See Figure 1: DOgSS Project Gantt Chart]. Due to COVID-19-related and other delays, our project was delayed, with the Gantt Chart reflecting our restart. Some of Phase 1 (obtaining ethics and governance approvals/amendments and project set-up), which had been initiated before the COVID-19 epidemic, was carried over from 2020 to 2022 (when further updates were undertaken); the project is currently in Phase 2 Cycle 2, with the protocol reported here from 2022 for Phase 1 and onwards (see Figure 1).

Patients would be recruited during Phase 2, which consists of three iterative cycles of planning, implementation, evaluation, and reflection [58]. The participants to be recruited are patients in the Stroke Unit at the RAH and the cycles with dogs visiting the Stroke Ward and data collection performed, as shown in Table 1. The interaction with each patient subsequent to consent, i.e., the dog visit and completing the pre- and post-visit surveys, can last up to approximately 15 min. The dog visiting can include patting, stroking, cuddling, and talking to/about the dog as led by the patient, and up to 15 min was estimated to accommodate the patient and allow for hand hygiene, data collection, patient questions, and also in keeping with veterinary advice (SJH, Associate Professor Animal Behaviour, Welfare and Ethics, School of Animal and Veterinary Sciences, University of Adelaide). Estimates are written broadly and permit flexibility with recruitment numbers at the time of a visit, allowing for patients needing to be taken away for imaging, for example, during the visit, a shorter visit if that is the patient’s request, and subsequently more patients to experience the dog-visit on a day and therefore fewer days of dog visits/cycle.

### 3.5. Participant Consent

Written consent on a Participant Consent Form will be obtained the day before or on the day of dog visitation, prior to the dog entering the room of the stroke patient. Participants will be informed verbally (as well as in the participant information sheet and consent form) that whether or not they participate in the study has no effect on their medical treatment. They can withdraw consent at any time during the interaction with the dog. However, if a participant wishes to withdraw consent after the day’s session has finished, the data may already have been de-identified, and it may not be able to be destroyed. Staff, informal supports, volunteers, and the dog handler will also be able to provide anonymous comments (Section A.4) through a sealed anonymous Comments Box situated in the hospital Stroke Unit; by providing comments, they consent to allowing their comments to be used in this study. This is a single-visit study for most patients, who will not be approached a second time. Where a patient has a subsequent dog visit, they will be re-consented using the ‘DOgSS Participant Information Sheet and Consent Form’, the latter to be signed again. Only patients able to give consent or who have a delegated representative able to give consent on their behalf will be able to be included in the study.

### 3.6. Risk and Risk Mitigation

#### 3.6.1. Potential Risks

It is expected that this study will not pose any substantial risks to participants. However, physical risks that could occur include the visiting dog physically harming participants or zoonotic illness [61,62,63,64,65,66,67], and emotional risks may include patients who may not feel comfortable being visited by the visiting dog or invoking feelings of distress upon exposure to the dog (e.g., missing their pet or formerly owned a pet but were forced to give up said pet as a result of a stroke). Patients who have had a stroke may also experience cognitive impairments and, thus, increased levels of emotional susceptibility [68].

#### 3.6.2. Strategies to Reduce Risk

Although risks are minimal, to mitigate these, close collaboration with the nurses at the Stroke Unit will be necessary to assist in excluding patients who may be unduly distressed or are substantially cognitively impaired. Participants will be asked prior to the dog entering the room whether they would like to participate. Participants who do not feel comfortable around the dog will not be recruited into the study. Furthermore, the visiting dog will be a therapy dog and thus trained to behave appropriately around vulnerable people.

If the participant becomes distressed due to the visit or does not want the dog to leave, there is nursing, medical, psychology, and social work support in the Stroke Unit for stroke and any participants who become distressed, and this support can be called upon by the nurse in charge and/or medical staff at any time. A flyer will also be displayed in the Stroke Unit and Neurology Ward as well as in lifts and corridors where the dog will walk, and notification will be made to hospital staff through electronic information for the workforce, to alert those along the way who may be concerned about the presence of/interacting with a dog (or wanting to interact with the dog), to give them the opportunity to avoid doing this. This also prevents anyone from outside of our participants or identified in our research as able to give anonymous feedback from interacting with the dog during our Action Research and the dog’s time at the hospital. It states:

“DOgSS Dogs Offering Support after Stroke is a research project * running on this ward. A trained dog and handler will be visiting to undertake research on the effects of a visiting dog on patients’ mood. Understanding and improving mood is important as how we feel is proven to enable other treatments to work more effectively. Please feel free to smile at the dogs but avoid direct contact with them as they are working”.* Animal Ethics Approval: Organization name and ethics approval number; Human Research Ethics Approval: Organization name and ethics approval number (Organization name and Animal Ethics Notification Reference Number).

The visiting dog will also have undergone regular preventative treatment for fleas and intestinal worms, minimizing the risk of zoonotic illness. Per the Volunteer Expression of Interest and Role Description, the hospital volunteer will assist with the patient room set up (e.g., towel/sheet on the bed) and help participants and those involved in the research project follow ward hand-hygiene and other infection-prevention protocols and procedures before and after interacting with the dog, and each Stroke Unit room has a basin for further hand washing after touching the dog and as appropriate. The dog handler will also have been trained to identify signs that may indicate the visiting dog may pose a risk to the physical safety of the participants. Although the risk of this is low, should this occur, the dog handler will remove the dog from the situation. Furthermore, if the dog is showing signs of stress, the handler will remove the dog to a quiet part of the ward or from the ward altogether.

#### 3.6.3. Planned Benefit

With respect to translation and subsequent organizational decision-making, implementation of a dog visiting service would be seamless, as all the processes would be in place at the end of the project, having been planned, carried out, assessed, and changed where necessary for optimal service provision.

### 3.7. Data Management and Analysis

A data management plan (DMP) was developed. All quantitative data (Likert Scale [52] and Smiley Faces [59]) will be analyzed using the McNemar and the Wilcoxon signed-rank statistical tests, respectively. Statistical significance will be set at an alpha level *p* < 0.05. Population descriptive variables will be reported as means ± standard deviation. Further statistical testing in conjunction with a statistician’s advice may be undertaken at the end of phase 2. Biomarker and ethogram analyses will be undertaken through a data analysis service provided by the University of Adelaide. Use will also be made of qualitative data such as project team reports to meetings to report on the decision-making processes undertaken in the Action Research project.

### 3.8. Plans for Dissemination and Publication of Project Outcomes

The overall results of the study will be communicated to the general public through public channels such as newsletters from The Hospital Research Foundation website and community-based stroke organizations. All data and quotes will also be in de-identified form to ensure participation in the study remains confidential. Reports are provided to our health service organization, CALHN, the University of Adelaide, as well as the grant providers, The Hospital Research Foundation (THRF). A research manuscript will also be written with the intention of publication in an academic journal. Conference presentations will be undertaken throughout and following the duration of this project.

### 3.9. Project Closure Processes

Accounts will be closed, and a final report will be made to the CALHN and University of Adelaide Research Offices, as well as The Hospital Research Foundation, on completion of the project, and both the human data and dog saliva samples will be managed per the DOgSS Data Management Plan and DOgSS Sample Management Plan, respectively, the latter as per the Animal Research Ethics Committee approval. De-identified data from this project may be utilized in future projects, publications, and/or conference presentations.

## 4. Results

The most relevant protocol documents for project planning, development, and initiation are provided, with our main checklists included below. Six checklist Tables are provided, which we used to assist with completing these activities in a timely manner: these are the Human Research Ethics Application (HREA)—Table 2, Animal Research Ethics Application (AREA)—Table 3, Governance and Finance—Table A1 (Section A.6), and Other Relevant Documentation and Activities—Table A2 (Section A.7), all of which can be extended to include columns for version number, submission date, approval date and expiry date. The Presentations and Publications—Table A3 (Section A.8) lists those to date, and the Operations folder—Table A4 (Section A.9), is a preparation list with accompanying documents for the Stroke Unit visit.

The online multi-question application (Table 2) includes the following:Project overview: team (CIs, AIs, Project Managers (PM(s)), disclosure of interests, restrictions, evaluations, location.Methods: participants, method specific questions, action interview and textual analysis research, participant specific questions.Recruitment: general, Action Research, consent, risk, benefit, data and privacy, attached relevant documents (see above).Selected relevant Human Research Ethics Committee (HREC) and review pathway (risk-related) and review scheme.Declarations from all researchers of their acceptance of the project protocol and endorsement from the Head of Neurology.The Project Description (PD) (Table 2) includes the following:Title: project team roles and responsibilities, resources necessary, funding required/secured.Background: literature review, rationale/justification, research question/aims/objectives/hypothesis, expected outcomes.Project design: setting, methodological approach, participants inclusion and exclusion criteria, sample size and statistical or power issues, recruitment strategies and timeframes.Approaches to provision of information to participants and/or consent, research activities, data collection/gathering, risk, and risk mitigation, data management and analysis; results, outcomes.Future plans: plans for return of research results to participants and for dissemination and publication of project outcomes, other potential data uses at project’s end, project closure process, plans for sharing and/or future use of data and/or follow-up research.

The offline Research Amendment (Table 2) includes a cover letter/email and a Research Amendment Request Form, which includes an amendment type, an overview of and reason for changes, participating sites, amendment documents (see below), and declaration.

Additional activities included sourcing, interviewing, and discussing service provision options with potential dog-handling services, e.g., a collaborative research agreement, a memorandum of understanding (MOU), or an agreement, before selecting a dog-handling service and embedding the selected arrangement in the project. The online ARE multi-question application in Table 3 includes Project Investigators, Approval Details, Project Overview, Participants and Potential Risk, Project Location and Funding, Indemnification Details, Animal Ethics Expertise Report, and External Investigator Declarations. The dog-handling service needs to use trained therapy dogs for this research and have the relevant workplace health and safety requirements in place. Included in the online ARE Amendment Request are the investigator and other personnel changes, general overview and project classification, ethical considerations, animal details, procedure details, e.g., protocol amendments, risk management, supporting information, investigator(s) declaration, and submission.

Table A1 in Section A.6 provides a template of governance documents, including legal requirements, the Site-Specific Application (SSA) for the RAH, Police Vulnerable People Check, CALHN Finance—Cost Centre, Progress Reports, and Approval Expiry Extensions. Other documents and activities undertaken by, e.g., Finance staff include their developing together with the Chief Investigator, the Budget Build developed from the Funding Agreement for how the finances are budgeted for, and also their Terms of Reference. These are not included in Table A1.

Other relevant activities and documentation, namely Human Resource documents including Job Descriptions, Training, and Meetings, are listed in Table A2 in Section A.7.

Presentations and Publications to date are provided in Table A3 in Section A.8; these started during Phase 1 and will continue through Phases 2 and 3. As important indicators of productivity, these should be referenced during grant reporting and possible future grant applications.

The Operations folder documents (and numbers prepared) for each visit to accompany the researcher and volunteers to the Stroke Unit are provided in Table A4 in Section A.9. Separate from the documentation, a small ice-box with ice and, in the research bag, three salivettes (two for each visitation, and one spare in case of a mishap) are carried for collection of saliva from the dog before and after the visitation.

## 5. Discussion

Acute stroke can have devastating emotional impacts on patients and close companions [52,53]. Animal Assisted Intervention has been shown to improve stroke-affected patients’ mood, enhancing quality of life [69]. Staff working in stroke care are also vulnerable to burnout, hence a need to identify means of reducing their stress [10,20,21,22]. Finding means of reducing the inherent stress associated with post-stroke experiences is crucial to maximizing the benefits of hospital interventions that may be impeded by the emotional experiences of patients. Our project will introduce regular dog visits to the Stroke Unit at the Royal Adelaide Hospital, adding to the international body of knowledge on the systemic impacts across both human and animal participants. We take into consideration not only our patients who have had a stroke but also their carers, family, and friends present at the time of the dog visit as they are key stakeholders in the lives of stroke patients, as well as relevant staff, volunteers, the dog handler and, in keeping with One Welfare [8] the dog itself. While trained therapy dogs are provided, the service is that of dog visiting, where, for example, the dog can be patted, stroked, cuddled, and talked about. DOgSS is seeking to collect ethographic and biomarker data from the non-human partner in the research team—the dog—as the wellbeing of all needs to be considered in keeping with One Welfare [70].

The Action Research study is planned as three cycles, with each stage reflecting, learning, and building on the previous stage [58]. Results will provide information as to the systemic impacts of a visiting dog on patients, visitors (informal patient supports), staff, and other key persons or areas in an acute hospital setting, as well as the physiological impacts on the dog(s) involved. With respect to translation and subsequent organizational decision-making, implementation of a routine dog visiting service to the organization would be seamless, as all the processes would be in place at the end of the project, having been planned, carried out, assessed, and changed where necessary for optimal service provision. The data from the project will also further current knowledge and understanding of AAI research.

We have described our protocol and included the most relevant documents for use or adaptation by other researchers in their consideration of undertaking such research. However, for completeness, it should be noted that prior to and surrounding the submission of a funding application, there is also activity that may not be planned for and which we have documented (Section A.1). While research publications include the research method, project development, and management is an essential part of research and yet not regularly addressed. Projects may not be initiated or run successfully without formal project planning, budgeting, and management, particularly when extraneous issues such as the challenges posed by the COVID-19 epidemic result in residual work for those who attempt to deal with the additional challenges arising. Project management needs to assess not only research risk but also project management risk and budget accordingly or risk worthwhile projects foundering or being abandoned at any phase of the undertaking.

## 6. Conclusions

DOgSS is a Royal Adelaide Hospital Stroke Unit Action Research study to find out if dog visits make a difference in the expressed mood of stroke patients and their visitors and the impact on Stroke Unit staff and hospital volunteers, as well as monitoring the wellbeing of the visiting dog. Project development and management are critical to research and for projects to be initiated and run successfully, particularly when challenges such as those posed by the COVID-19 epidemic arise. We summarise the processes for grant funding application, including conception, funding application, ethics (human and animal) applications, governance and finance administration, as well as the Action Research method, which includes assessment and reflection, plus implementation adjustments to build on previous cycle action for enhanced understanding of multifaceted information. We provide checkpoint lists for project planning, on/offline submission and subsequent amendment, budgeting, management, and administration. This may assist researchers when internal project management software is not available to them and/or they are not aware of the complexities of undertaking such multi- and inter-disciplinary multifaceted research. Relevant project management and research documents are also included for information and can be used as templates for assistance in the development of such documents. As project processes vary according to the jurisdiction, our protocol would not necessarily be transferrable but also offers insight into how labor-intensive and lengthy the process can be. As such, our protocol can be used for information or adapted by other organizations to support health promotion internationally in one of the five important action areas of the Ottawa Charter, namely that of reorienting health services.

## Figures and Tables

**Figure 1 ijerph-20-06780-f001:**
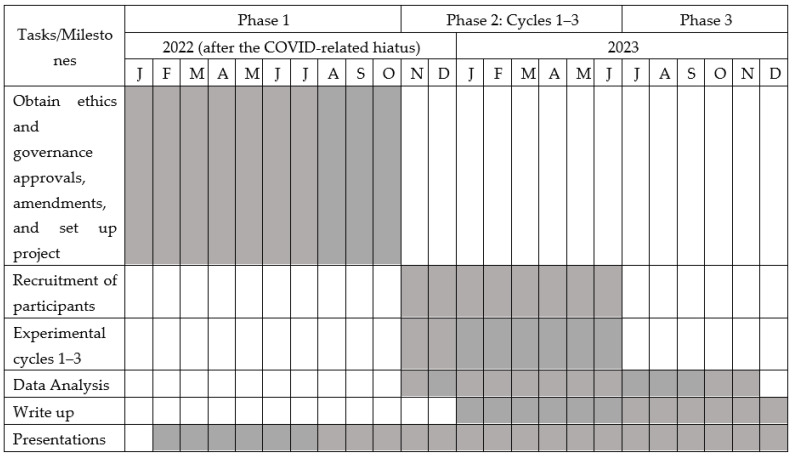
DOgSS Project Gantt Chart showing the anticipated three phases of the project (with projected timing in color), with flexibility required for any Action Research and other changes.

**Table 1 ijerph-20-06780-t001:** Three phases of the project, with Phase 2 including the three data collection cycles.

Phases	Details		
Phase 1: Planning	Human and animal ethics write-up—approval to be obtained before commencing Phase 2. Implementation interrupted (COVID-19, etc., related), and ethics and governance amendments and new agreements required, as well as sourcing a different dog visiting service provider, causing extension of this phase. Project administration, coordination, and management, including planning.		
Phase 2: Data collection	Cycle 1	1 visit/week for up to 5–6 weeks (*n* = approximately 20 patients, although this may be more, depending on the number of patients seen per day and consideration of the welfare of the dog), then 1 month or more of no visits, dependent on phase data analysis, ethics amendments, etc.	Data to be collected each visit: Pre-visit: hospital volunteer/researcher to administer Likert Scale [52] and Pick-a-Mood smiley faces to stroke patients [59], * dog handler to collect salivary sample from visiting dog. During visit: hospital volunteer/researcher to collect ethogram [60], i.e., observational data on the dog (Section A.5). Post-visit: hospital volunteer/researcher to administer Likert Scale and Pick-a-Mood smiley faces to stroke patients. Informal patient supports, as well as Stroke Unit staff, volunteers, and the dog handler, will be able to provide anonymous comment in Cycle 1.
	Cycle 2	1 visit/week for up to 5–6 weeks (*n* = approximately 20 patients, although this may be more, depending on the number of patients seen per day and consideration of the welfare of the dog), 1 month of no visits, dependent on phase data analysis, ethics amendments, etc.	Cycle 2 will follow Cycle 1 after adjustments made as required, based on Cycle 1 results. Additionally, the mood tools will be used with consenting informal supports, subject to ethics approval before this cycle.
	Cycle 3	1 visit/week for up to 5–6 weeks (*n* = approximately 20 patients, although this may be more, depending on the number of patients seen per day and consideration of the welfare of the dog).	Process to be adjusted based on outcomes of Cycles 1 and 2. Dependent upon the dog visiting contract, a further 5–6 weeks (*n* = 20 patients) may be added to this cycle, thereby also catering for up to approximately 20 patients to have a second dog visit during Phase 2. Final numbers and timing will also be subject to COVID-19 constraints, which may also impact the dog visiting contract.
Phase 3: Analysis and write up	Data analysis. Manuscript to be written and submitted for publication, likewise for PowerPoint and other presentations.		

* The dog handler will collect a salivary sample from the dog just prior to the first patient visit and just after the last patient visit on the day of the visit and samples taken will be held in a container with ice for transport by the Chief Investigator/Project Team Member to the University of Adelaide for storage and subsequent analysis of stress biomarkers (Cortisol and IgA) [37].

**Table 2 ijerph-20-06780-t002:** Human Research Ethics Application (HREA) to the Human Research Ethics Committee.

Low to Negligible Risk (LNR)	HREA, Amendment, and Related Documents
HREA	Online Multi-question HREA
HREA Attachments	Ethics Covering Letter
	CVs: all Project Researchers and Personnel (includes their expertise relevant to the research activity)
	Volunteer Expression of Interest (VEOI)
	Volunteer Role Description (VRD)
	Dog-handler Sign-in Sheet
	Participant Information Sheet and Consent Form (PIC)
	Animal Research Ethics (ARE) Approval Certificate
	Data Management Plan (DMP)
	Project Description (PD)
	Likert Mood Scale
	‘Smiley’ Faces selection
	Anonymous Comments Form
	Emailed acceptances from all researchers
	Head of Department Endorsement
HREA Outcome	Research Governance Officer (RGO) Authorisation Letter (HRE Committee Approval)
HRE Amendment	Covering letter/email. Offline Research Amendment Request Form
Attachments	CVs: new Project Personnel
	VEOI—tracked (showing edits) and clean copies
	VRD—tracked and clean copies
	PIC—tracked and clean copies
	Poster/Flyer to inform staff of dog visit
	PD—tracked and clean copies
Notification to UoA of External HREA Approval(s)	Online multi-question application
Notification	Approval documentation
Notification to UniSA of External HREA Approval(s)	UniSA HREC sent copies of ethics updates as received
Notification	Covering email
	Approval documentation

**Table 3 ijerph-20-06780-t003:** Animal Research Ethics (ARE) Application (AREA).

Category	UoA AREA, Amendment, and Related Documents
AREA	Online multi-question application
AREA Attachments	Data Management Plan (DMP)
	Sample Management Plan (SMP)
	Evidence of animal ethics training (CIs, AIs, PMs)
	Researcher (Medical) Certificate of Practice
	Use of privately owned animals consent form
	External Investigator Declaration
ARE Amendments	Online amendment request summary and justification
	Application outcome details and ARE Approval certificate

## Data Availability

Not applicable.

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
