# Peer review of "Developing and Planning a Protocol for Implementing Health Promoting Animal Assisted Interventions (AAI) in a Tertiary Health Setting"

_ijerph, 2023, doi:10.3390/ijerph20186780_

Round 1

Reviewer 1 Report

The protocol presented in this manuscript provides rich background information for researchers interested in assessing animal-assisted interventions (AAI) in hospital settings in the future. The authors focus particularly on project management and operations, which are often overlooked in this field of research, despite their clear practical importance and challenges. The protocol is well described, and the text is well written; thus, minor changes are suggested. More information about the dog visit per se is recommended, as this is an important aspect of their future work, and various comments were made to improve the readability of the text.

TITLE

The first part of the title is very vague "A way forward in health service reorientation". It could be removed as the title is long.

ABSTRACT

Objectives number 2 and 3 are not very clear in the abstract - "informal patient supports?"; "undertaking and reporting on the AR?". Is the latter objective 4? I suggest the sentence is rephrased. These are much clearer in the Introduction of the paper (lines 113-126).

BACKGROUND

- Line 53: please rephrase this to improve coherence/clarity: "the usefulness is not always clear [22-24]". It seems contradictory as the authors just mentioned various benefits of AAI.

- Line 55: in animals

- Line 74: remove the year - "2022"

MATERIAL AND METHODS

- Lines 150-153: please rephrase to improve readability. It is confusing due to its length and multiple uses of "etc". Brackets might help.

- Line 207: You could give examples of potential changes.

- Line 226: Perhaps move this paragraph up and the previous paragraph down. It is strange/confusing to read about the selection of participants before understanding the criteria for it.

- Line 255: Remove the "etc" or explain it further; it is not clear what the "etc" is about.

- Figure 1: Could you add the cycles to this figure to facilitate understanding? It is hard to estimate when the cycles start/end.

Could you change the font in Figure 1 to match the font used in the main text?

What will happen between the dog and the patient during the visits? That is not clear in the main text or Figure 1. Free interactions led by the patient? Similar standard protocol followed in all sessions? Various studies (e.g., those conducted by Barcelos) indicate that different dog-related activities have different well-being outcomes in humans, thus, it is important to consider this.

- Line 266: any particular reason for this limit of 15 minutes? Consider adding the rationale.

- Line 290: wouldn't it be better to not include these second sessions in the study? You already mentioned that reaching 80 participants is not likely to be an issue; it would make more sense to not have repeated patients in your sample.

- Line 335: could you please specify here or previously (e.g., line 310) in the manuscript which attributes were used to classify this dog as a therapy dog?

- Line 343: consider adding a sentence with a focus on the welfare of the dog - e.g., giving a break/stopping the visit in case the observer is concerned about the dog showing signs of stress. A resting area for the dog will be provided in case it wants to move away from the interaction, etc.

- Line 347: can you describe here the variables to illustrate this better?

- Line 352: same comment as above.

- Line 370 and line 377: the headings 3.9 and 4. are not formatted appropriately - the dot is misplaced.

- Table 1: I suggest you emphasise more in the text that Table 1 contains data collection information. The heading of this section is focused on participant recruitment and timeframes, but data collection is a very important aspect of your project.

RESULTS:

General comments:

- I wonder if Tables 2 to Table 6 couldn't be moved to the Appendix and just a summary with a few examples should be provided in the main text?! You could add headings to the Results section to improve clarity.

- Line 387: should you close the bracket after "Table 2"? This paragraph is very hard to read due to the number of brackets, colons and semicolons used and things being listed. Consider rephrasing it to improve clarity, e.g., 2-3 sentences instead of a very long one with numerous of punctuation marks.

- Line 395: same comment as above; one-sentence paragraph is not ideal - hard readability. There is a typo at the end of the paragraph: two dots.

- Table 3: is there a typo here "AE Approval"? Should it be "ARE"?

DISCUSSION:

- Line 478: why do you mention the implementation of another service? This seems a bit out of context; perhaps you could introduce this topic better.

APPENDIX 2:

- The way the times are presented should be consistent, e.g., always in this format 9:30 or this 9.30.

- "Meet with NUM" - please explain what NUM is.

- Arrows typically mean that one thing is the outcome of another, or subsequent to another. In the workflow, you could use different shapes (e.g., rectangles) if subsequence/outcome was not meant. The idea of the workflow is good, but the design/shapes could be improved.

Author Response

Please see attached response.

Reviewer 2 Report

Thank you for the opportunity to review your paper. This sounds like a very interesting study, and I look forward to reading the results once complete. This is a well written paper, however I have difficulty understanding how this protocol holds relevance for researchers outside this specific project. As a protocol, it seems to largely describe the initial administrative work completed to run a research project which will differ between organisations. It would instead be beneficial to understand what the visitation sessions would look like with a focus on the human participants and the dogs. For example, for the human participants, how long will the sessions run for, where will the visits take place, how will participants interact with the dog, who is leading the visits, is there a specific structure that these visits need to follow, etc? For the therapy dogs included, how many days per week are the dogs working, how many visits will they do each day, what specific training will the dogs need to complete to be eligible to participate, what is the inclusion criteria for the human-canine visitation teams, what measures will be taken to support the welfare of the dogs, will there be a control saliva sample of the dogs taken on a day that they  are not working? at what time will this sample be taken? A protocol of how the DOgSS program is delivered would be beneficial to the readers. 

Author Response

Please see attached response.

Reviewer 3 Report

The authors report a plan to develop a study of an animal-assisted intervention for patients in an acute hospital setting who have experienced a stroke. 

General comments

This study proposed is interesting.  However, the paper is long and includes a lot of detail, which results in a paper that is not reader friendly.  I think the paper could shortened and editing to be more reader friendly.

The authors provide a literature review, but the paper would benefit from a tighter linking of both the human stroke recovery literature and the AAI literature to the study design. What gaps in the existing literature will this study address?  The outcome measures to be used for humans and animals are not clearly described, justified or linked to science.

If this study is conducted - what would next steps and/or follow-up studies.

Introduction

Section 1 clarify the connection between the Ottawa Charter and AAI.

Line 76:  The authors state, “, but there is no right to public access.”  Please clarify what this means.  Public access to what?

Line 113: Define “action research.” 

Materials and methods

Clarify what human and animal outcome measures are being used and why.

Line 129 & 130: It is unclear why the authors insert the phrases ‘on the business’,  and ‘in the business’.  Why not just say “project management” and “operations.”

How was the duration of the dog visits (15 minutes) determined?

Provide more information about why a power analysis was not conducted.

Reviewer 4 Report

I would like to commend the authors for approaching this topic and for preparing this manuscript. This novel study of developing an AR framework for AAI in a tertiary health setting is timely, welcomed and critical in pursuing wider acceptance and utilization of the valuable resources that AAI provide for various patients. 

My one suggestion regards the zoonoses mitigation part of protocol. Given the health benefits of AAI and the vulnerability of patients in the rehabilitation phase after a stroke, as well as concerns for zoonoses from patients and healthcare workers, well outlined infection control measures must be included in any AR framework for AAI.  While the risk is minimal, the addition of a veterinarian to the research team can prove to be a helpful and valuable resource, considering their wealth of knowledge and experience. 

Congratulations on a well done study and manuscript. 

Round 2

Reviewer 3 Report

The authors have adequately addressed my recommendations.